# Gelation Process Optimization of Shrimp Surimi Induced by Dense Phase Carbon Dioxide and Quality Evaluation of Gel

**DOI:** 10.3390/foods11233807

**Published:** 2022-11-25

**Authors:** Ouyang Zheng, Qinxiu Sun, Andi Dong, Zongyuan Han, Zefu Wang, Shuai Wei, Qiuyu Xia, Yang Liu, Hongwu Ji, Shucheng Liu

**Affiliations:** 1Guangdong Provincial Key Laboratory of Aquatic Products Processing and Safety, Guangdong Province Engineering Laboratory for Marine Biological Products, Guangdong Provincial Engineering Technology Research Center of Seafood, Key Laboratory of Advanced Processing of Aquatic Product of Guangdong Higher Education Institution, College of Food Science and Technology, Guangdong Ocean University, Zhanjiang 524088, China; 2Jiangsu Key Laboratory of Marine Bioresources and Environment, Jiangsu Ocean University, Lianyungang 222005, China; 3Collaborative Innovation Center of Seafood Deep Processing, Dalian Polytechnic University, Dalian 116034, China

**Keywords:** shrimp surimi, dense phase carbon dioxide, process optimization, gel properties

## Abstract

Dense phase carbon dioxide (DPCD) is a new non-thermal method to induce surimi gel. However, the gel quality is affected by many factors, such as DPCD treatment time, temperature, and pressure, which makes it complicated to determine its operating parameters. Box-Behnken and backward linear regression were used to optimize the conditions (temperature, pressure, and treatment time) of DPCD-induced shrimp surimi gel formation, and a model between shrimp surimi gel strength and treatment conditions was developed and validated in the present study. Meanwhile, the heat-induced method was used as a control to analyze the effect of DPCD on the quality of shrimp surimi gel in the present study. The results showed that DPCD treatment affected the strength of shrimp surimi gel significantly, and the pressure of DPCD had the greatest influence on the gel strength of shrimp surimi, followed by time and temperature. When the processing pressure was 30 MPa, the temperature was 55 °C, and the treatment time was 60 min, the gel strength of the shrimp surimi was as high as 197.35 N·mm, which was not significantly different from the simulated value of 198.28 N mm (*p* > 0.05). The results of the gel quality properties showed that, compared with the heat-induced method, DPCD reduced the nutrient and quality loss of the shrimp surimi gel, and increased the gel strength and gel water-holding capacity. The results of low-field nuclear magnet resonance showed that DPCD increased the binding capacity of shrimp surimi to bound water and immobilized water, and reduced their losses. Gel microstructure further demonstrated that DPCD could improve shrimp surimi gelation properties, characterized by a finer and uniformly dense gel network structure. In summary, DPCD is a potential method for inducing shrimp surimi to form a suitable gel. The prediction model established in this study between DPCD treatment temperature, pressure, time, and gel strength can provide a reference for the production of shrimp surimi by DPCD.

## 1. Introduction

White shrimp (*litopenaeus vannamei*) is one of the three largest species of shrimp in the world. It has the characteristics of fast growth, strong disease resistance, and high yield [1]. Shrimp surimi is one of the deep-processed products of shrimp (a kind of surimi), including shrimp sausages, shrimp meatballs, and other shrimp surimi products. The traditional method of producing surimi products is mainly heat induction (water boiling or steam), but water boiling is easy to cause the loss of water-soluble nutrients, and too high temperature is easy to cause the loss of heat-sensitive nutrients [2]. In addition, because the traditional heating rate and heat transfer rate are relatively slow, the residence time at 50–70 °C during warming is too long, and the degradation of myosin by some high-temperature-resistant endogenous proteases (50–70 °C optimum temperature) leads to the deterioration of surimi gel [3], which seriously restricts the promotion of surimi and the related industrial chain development. In order to solve the problem that traditional heat induction easily leads to the deterioration of the gel of surimi products, many researchers have adopted new processing methods to induce protein gelation. For example, ohmic heating [4], microwave heating [5], ultra-high pressure [6], dense phase carbon dioxide (DPCD), and other technologies have been used to induce gelation and improve its gel properties. DPCD refers to CO_2_ that is coupled with a certain pressure (<50 MPa) and temperature (<60 °C), mainly supercritical CO_2_ [7]. Sometimes gaseous and liquid carbon dioxide near critical points (31.1 °C and 7.38 MPa) are included. As a non-thermal processing technology (temperature below 60 °C), DPCD can effectively improve the gel properties of gel products, which provides a new technical approach for the development of protein gel products [8]. Meanwhile, DPCD is a green processing technology due to its safety and nontoxicity, environmental friendliness, residue-free, easy operation, and control [9]. Therefore, the DPCD technique is a very promising technology for surimi processing. At present, the DPCD technique has been successfully used to modify some gelation products, such as minced mutton [10], silk fibroin gel [11], and shark skin collagen gel [12]. The researchers found that DPCD treatment conditions (temperature, pressure, and treatment time) are important factors in determining the quality of gel, and the optimal treatment conditions for different food proteins are different. Our previous research found that DPCD could induce shrimp surimi to form gel [13]. However, the gel quality is affected by many factors, such as treatment time, temperature, and pressure, which makes it complicated to determine its operating parameters. Optimization is essential for technology development to get an acceptable product with desired quality parameters. The response surface methodology is a statistical tool for optimization in order to explore the change of gel strength of shrimp surimi during the DPCD treatment. Therefore, a Box-Behnken response surface methodology test design and regression analysis were used to study the quantitative between DPCD treatment pressure, temperature, time, and gel strength of shrimp surimi, and the processing parameters were optimized. Meanwhile, the gel properties (nutrient content, weight loss, sensory evaluation, gel strength, gel water-holding capacity, and gel microstructure) of shrimp surimi gels with DPCD were analyzed using heat-induced gels as a control. The results can provide theoretical guidance and technical reference for the green processing of surimi products.

## 2. Materials and Methods

### 2.1. Chemicals

Carbon dioxide (purity 99.99%) was purchased from Zhanjiang Oxygen Plant. Glutaraldehyde, sodium dihydrogen phosphate, disodium hydrogen phosphate, ethanol, isoamyl acetate, and tert-butanol were purchased from Guangdong Guanghua Sci-Tech Co., Ltd. (Guangzhou, China), and all reagents were analytically pure.

### 2.2. Preparation of Shrimp Surimi

Fresh living white shrimp (36–40 strips/kg) was purchased from Dongfeng aquatic market at Xiashan (Zhanjiang, China), and transported to the laboratory with water and oxygenation. After washing with tap water, uniform and intact shrimp were selected. After the shrimp were decapitated, shelled, and intestinal glands, the shrimp was minced with a knife, and then rinsed repeatedly three times (15 min/time) in ice water containing 5 times the mass of the shrimp. After squeezing and dehydration, the minced meat was added with 3% salt, and then chopped in an ice water bath for 15 min to make a shrimp surimi sample (water content of about 80%). The shrimp surimi was put into a customized mold and treated with different conditions (heat or different conditions of DPCD) immediately.

### 2.3. Treatment of Shrimp Surimi with DPCD

Figure 1 shows the working diagram of DPCD processing equipment. The refrigeration and refrigeration cycle was first opened, and the required temperature of the treatment tank was set; then, the mold with the shrimp surimi sample was put into the treatment kettle, which was subsequently sealed. The high-pressure pump was opened to pump CO_2_ gas. The high-pressure pump was shut down when the pressure rose to the required pressure. Then, the valves in and out of the treatment kettle were sealed to maintain the required pressure and temperature in the treatment kettle. After a period of static treatment, CO_2_ was released, and the air pressure was reduced to normal pressure. Finally, the sample was taken out, and treatment was completed. The processed samples were cooled to room temperature rapidly and then placed at 4 °C for 12 h. The samples were taken out from the refrigerator at 4 °C and placed at room temperature for 1 h before the test to equilibrate the sample temperature to room temperature. Shrimp surimi before DPCD treatment and after DPCD treatment are shown in Figure 2.

### 2.4. Treatment of Shrimp Surimi with Heat

Two-stage heating was utilized to prepare shrimp surimi products. Briefly, shrimp surimi was first heated in a 40 °C water bath for 30 min, and then in a 90 °C water bath for 30 min. After the heating, the surimi products were quickly cooled with ice water and then placed at 4 °C for 12 h. The gel samples were balanced to room temperature before the test.

### 2.5. Determination of Gel Strength

The determination of gel strength refers to the method of Sun, Chen, Xia, Kong, and Diao [14]. The breaking strength and breaking distance of the shrimp surimi gel were determined by a TMS-PRO-type texture analyzer with a *p*/_0.5_ probe. Gel strength was measured using a puncturing mode with a test speed of 1 mm/s and a depth of 10 mm punctured into the gel. The formula for gel strength was as follow:Gel strength (N mm) = breaking strength (N) × breaking distance (mm)(1)

### 2.6. Design of Experiments

According to the permissible pressure, temperature range of DPCD treatment, and our previous experimental results [15,16,17], the qualitative law between the pressure, temperature, time of DPCD treatment, and the gel strength of shrimp surimi was studied by using double-factor equal repeated experimental design. The factors and levels were set as follows: (1) the effect of DPCD pressure on the gel strength of shrimp surimi: the fixed treatment time was 20 min, the treatment temperature was 50 °C and 60 °C, and the treatment pressure was 0.1, 5, 10, 15, 20, 25, and 30 MPa; (2) the effect of DPCD temperature on the gel strength of shrimp surimi: the fixed treatment time was 20 min, the treatment pressure was 20 MPa and 25 MPa, and the treatment temperature was 35, 40, 45, 50, 55, and 60 °C; (3) The effect of DPCD treatment time on the gel strength of shrimp surimi: the fixed treatment temperature was 60 °C, the treatment pressure was 20 MPa and 30 MPa, and the treatment time was 10, 20, 30, 40, 50, and 60 min, respectively.

Taking the gel strength (*y*) of shrimp surimi as the test response and the treatment pressure (*z*_1_), temperature (*z*_2_) and time (*z*_3_) as the experimental factors, the relationship between the factors of the DPCD and the gel strength of the shrimp surimi was studied by Box-Behnken design and regression analysis, so as to optimize the gelation process of shrimp surimi induced by DPCD. The factor levels and coded values for the DPCD treatment were set (Table 1) based on the results of above. The levels of the factors in Table 1 were coded because the units and the change range of each factor were different, and the effect of the unit could be eliminated after coding, and the range of change of each variable was [–1, 1]. In the regression analysis, the importance of the influence can be directly judged by the coefficient of the variable.

### 2.7. Determination of Nutritional Component

Evaluation of the ash content was carried out by using HYP-Ⅱ muffle furnace (Shanghai Fiber Testing Instrument Co., Ltd., Shanghai, China), following the National Standards of the People’s Republic of China method No. GB/T 5009.4-2016. Evaluation of the total protein content was carried out by using an apodest Kjeldahl nitrogen analyzer (Gerhardt, Konigswinter, Germany) following the People’s Republic of China method No. GB/T5009.5-2010. Evaluation of the moisture content carried out by using drying oven HB-SY-4B (Suzhou Humon Oven Manufacturing Co., Ltd., Suzhou, China) following the People’s Republic of China method No. GB/T 5009.3-2010.

### 2.8. Sensory Evaluation

Sensory evaluation was referred to the method of Balti, Mansour, Zayou, Balc’h, Brodu, and Arhaliass et al. [18] with slight modifications. The samples were cut into 5 mm-thick slices and randomly coded, and sensory evaluations were performed by 10 trained sensory reviewers according to the criteria in Table 2. Evaluation indexes included odor, color, tissue morphology, and taste. Each item was assigned a score range of 1–5 points, with each weighing 0.1 for odor, 0.1 for color, 0.5 for tissue status, and 0.3 for taste. The overall score was calculated by the following formula:Total score = (*i* = 1, 2, 3...*n*)(2)
where *a* is the evaluation index, and *b* is the evaluation weight. The scoring results were expressed as the mean ± standard deviation of the data obtained from 10 evaluators. The higher the score, the better the quality of shrimp surimi.

### 2.9. Determination of Weight Loss and Water-Holding Capacity

The determination of weight loss was according to the method of Xiao, Xin, Wei, Feng, and Liu [19]. The weight of fresh shrimp surimi was weighed *m*_1_ (g), and after inducing gel formation by different methods, the surface water of the shrimp surimi was dried with filter paper, and then the weight of the shrimp surimi was accurately weighed (*m*_2_, g). The weight loss was calculated according to the following formula:(3)$$\mathrm{Weight}\;\mathrm{loss}\;(\%)\;=\frac{m_{1}-m_{2}}{m_{1}}\times 100$$

The water-holding capacity of the samples was measured as described by Liu et al. [20]. The gel sample with a certain weight of *ω*_1_ (g) shrimp surimi was wrapped in three layers of filter paper and placed in a centrifuge tube. Then the sample was centrifuged at 8000× *g* for 10 min. After centrifugation, the filter paper was removed and the weight of the gel sample was measured as *ω*_2_ (g). The water-holding capacity of the gel was calculated by the following formula:(4)$$\mathrm{Water}\;\mathrm{holding}\;\mathrm{capacity}\;(\%)\;=\frac{\omega_{2}}{\omega_{1}}\times 100$$

### 2.10. Determination of Water Distribution of Shrimp Surimi

The moisture distribution of shrimp surimi was determined by an NMI20-060H-I low-field nuclear magnetic resonance (LF-NMR) apparatus (Niumag Electric Corporation, Shanghai, China). The proton resonance frequency was 21 MHz, and the magnet intensity was 0.43 T. After normalizing the original data, the Caar–Purcell–Meiboom–Gill pulse sequence and the CONTIN algorithm were used to calculate the transverse relaxation time *T*_2_.

### 2.11. Texture Profile Analysis (TPA) of Shrimp Surimi

The texture properties of the shrimp surimi samples were analyzed by TPA using a texture analyzer (TMS-Pro, FTC, Sterling, VA, USA) according to the method described by Pan et al. [1]. The detection indexes included hardness, adhesiveness, cohesiveness, chewiness, and springiness. The instrument was calibrated with a 5 kg load cell, and the test probe was a round cake with a diameter of 50 mm. Sample test parameters were as follows: the height of the probe rising to the sample surface was 30 mm, the compression deformation was 50%, the detection speed was 60 mm/min, and the trigger force was 0.5 N.

### 2.12. The Color of Shrimp Surimi

The values of *L**(brightness), *a** (redness), and *b** (yellowness) of the sample were measured with a ZE-6000 chromaticity meter (Juki Company, Tokyo, Japan) at room temperature. The color aberrator was calibrated with the white standard version (*L** = 95.26, *a** = 0.89, *b** = 1.18) before determination.

### 2.13. Microstructure of Shrimp Surimi Gel

The microstructure of shrimp surimi was observed by a scanning electron microscope (SEM) (S3400, Hitachi, Tokyo, Japan) as described by Sun, Kong, Liu, Zheng, and Zhang [21], with little modification. Shrimp surimi samples were cut to a size of 2 × 2 × 5 mm^3^ small pieces and fixed in 2.5% glutaraldehyde solution for 12 h at 4 °C. Then, the fixed samples were washed three times for 10 min each with 0.1 mol/L phosphate buffer (pH 7.2). A gradient of ethanol was added to dehydrate the samples in the following order: 30%, 50%, 70%, 90%, and 100% ethanol, each concentration twice, each dehydration 10–15 min. Then, the sample was soaked in a 1:1 mixture of isoamyl acetate and ethanol for 15 min and shaken appropriately. The mixed liquid was replaced with 100% tert-butyl alcohol and soaked for 15 min. Then the sample was freeze-dried in vacuum, sprayed with gold, and observed by SEM with a magnification of ×1500.

### 2.14. Statistical Analysis

A total of three batches of experiments were carried out in this study, with three parallel samples in each batch. The data were expressed as mean ± standard deviation. Analysis of variance, Tukey’s HSD multiple comparisons (with a confidence interval of 95%), and response surface regression analysis were performed using JMP10.0 statistical software.

## 3. Results

### 3.1. Effect of DPCD Treatment Conditions on the Shrimp Surimi Gel

As can be seen in Figure 3, the shrimp surimi treated at 0.1 MPa CO*_2_* failed to form a better gel state due to the soft tissue, and the gel strength was not measured on the texture analyzer. This indicates that atmospheric pressure CO_2_ is unable to induce gel formation in shrimp surimi. Under the same treatment temperature, the gel strength of shrimp surimi significantly increased with the increase in treatment pressure (*p* < 0.05); that is, the gel strength of shrimp surimi was positively correlated with the treatment pressure. The treatment pressures of both DPCD at 50 °C and 60 °C for 20 min resulted in the formation of better morphological gels of shrimp surimi, which were firmer and more elastic as the treatment pressure increased. This is also consistent with the results of gel strength (Figure 4).

The treatment of shrimp surimi using DPCD at 35 °C and 40 °C failed to form a better gel state (Figure 3) due to the soft tissue, and the gel strength (Figure 4) was not measured on the texture analyzer. The gel strength of shrimp surimi increased significantly (*p* < 0.05) with the increase in DPCD treatment temperature in the range of 45–60 °C at the same pressure.

Under the fixed DPCD pressure and temperature, the strength of shrimp surimi gel increased with the extension of DPCD treatment time (Figure 4) (*p* < 0.05). The appearance of shrimp surimi became better, and the tactile elasticity became bigger (Figure 3). Combined with the above results and our previous research results, we confirmed the range of DPCD parameters that can form a relatively suitable gel (pressure: 10–30 MPa, temperature: 50–60 °C, time: 10–60 min) for the following the Box-Behnken design.

### 3.2. Regression Model of DPCD-Induced Shrimp Surimi Gel

Based on the Box-Behnken design and factor level coding table (Table 1), a test scheme for the gelation of shrimp surimi induced by DPCD was established (Table 3). Fifteen test numbers were randomized, and then the gel strength of DPCD-induced shrimp surimi gel was measured using a texture analyzer (Table 3). The regression model established by response surface regression analysis on the data in Table 3 was as follows:y = 164.85 + 26.05 *×* 1 + 20.52*x*_2_ + 23.29*x*_3_
*−* 17.07*x*_1_*x*_2_
*−* 6.89*x*_1_*x*_3_
*−* 7.75*x*_2_*x*_3_
*−* 10.28*x*_1_^2^
*−* 7.34*x*_2_^2^ + 2.33*x*_3_^2^(5)

Table 4 shows that the predicted *R*^2^ of 0.85 is in reasonable agreement with the Adjusted *R*^2^ of 0.97; i.e., the difference is less than 0.2. Adeq precision measures the signal-to-noise ratio. A ratio greater than 4 is desirable. The ratio of 26.46 indicates an adequate signal. This model can be used to navigate the design space.

The coefficient of regression model (5) was tested for significance, and the results are shown in Table 5. As can be seen from Table 5, the effect of *x*_32_ terms (i.e., quadratic term of treatment time) on the strength of shrimp surimi gels in the regression model was insignificant (*p* > 0.05). Therefore, the regression model needs to be further optimized by excluding the insignificant term in (5). After removing *x*_32_ from regression model (5) and then conducting response surface regression analysis on the data in Table 2, the regression model was established as follows:*y* = 166.28 + 26.05*x*_1_ + 20.52*x*_2_ + 23.29*x*_3_ − 17.07*x*_1_*x*_2_ − 6.89*x*_1_*x*_3_ − 7.75*x*_2_*x*_3_ − 10.46*x*_1_^2^ − 7.52*x*_2_^2^(6)

The coefficient of regression model (6) was tested for significance, and the results are shown in Table 5. As can be seen from Table 5, each item in the regression model has a significant impact on the gel strength of shrimp surimi (*p* < 0.05). Therefore, each item in Table 5 can be retained in the regression model (6).

Regression model (6) was used to predict the gel strength of shrimp surimi in the 15 tests in Table 2, and linear correlation analysis was conducted between the predicted value and the experimental value of shrimp surimi gel strength. The results are shown in Figure 5. It can be seen from Figure 5 that the predicted value of gel strength presented a suitable linear correlation with the experimental value, with determination coefficient *R*^2^ = 0.99 and root mean square error RMSE = 3.72. This indicates that regression model (6) could explain 99% of the changes in the gel strength caused by the mixing pressure, temperature, and time, and the regression model (6) had a high reference value. The analysis of variance and the lack of fit test of the regression model (6) were performed. As can be seen from Table 6, the regression model was significant (*p* < 0.05), while the loss of fit was not significant (*p* > 0.05), which indicates that the fitting of regression model (6) was feasible within the variation range of independent variables (−1, 1).

The results of the coefficient significance test, coefficient of determination (*R*^2^), analysis of variance, and loss-of-fit test of the regression model (Table 6) fully indicated that regression model (6) could be used to predict the gel strength of shrimp surimi under different mixed treatment pressure, temperature, and time and optimize the process parameters.

### 3.3. Change of Surimi Gel Strength during DPCD Treatment

Since the Box-Behnken test design encodes the pressure, temperature, and time factors for the DPCD treatment, eliminating the influence of each factor unit, they were all varied in the range [−1, 1]. Therefore, the gel strength of each shrimp surimi was calculated sequentially based on the absolute values of coefficients in regression model (6). According to the absolute values of each coefficient in Table 7, the order of magnitudes affected by the strength of each shrimp surimi gel was pressure once term (*x*_1_), time once term (*x*_3_), temperature once term (*x*_2_), pressure and temperature cross term (*x*_1_*x*_2_), pressure squared term (*x*_1_^2^), temperature and time cross term (*x*_2_*x*_3_), temperature squared term (*x*_2_^2^), and the intersection of pressure and time (*x*_1_*x*_3_). Where the pressure once term (*x*_1_), temperature once term (*x*_2_), and time once term (*x*_3_) were positively related to the gel strength of shrimp surimi, while the pressure to temperature cross term (*x*_1_*x*_2_), pressure squared term (*x*_2_), temperature to time cross term (*x*_2_*x*_3_), temperature squared term (*x*_3_), pressure to time cross term (*x*_1_*x*_3_) were negatively related to the gel strength of shrimp surimi.

Based on the regression model (6), the interaction among DPCD pressure, temperature, and time was analyzed (Figure 6). In the interaction figure, if any two factors are parallel to the two curves at the vertical and horizontal intersection, it indicates that there is no interaction between the two factors; if two curves are intersected, it indicates that there is an interaction between the two factors [22]. It can be seen from Figure 6 that both curves were crossed between any two factors, indicating that the interaction between pressure and temperature, pressure and time, and temperature and time of DPCD treatment on the strength of shrimp surimi gel were significant (*p* < 0.05). Therefore, the effect of DPCD treatment on gel strength was not caused by a single factor but by the combined action of temperature, pressure, and time.

In order to investigate the influencing of three factors (pressure *x*_1_, temperature *x*_2,_ and time *x*_3_) on the strength of shrimp surimi gels in regression model (6), one of them was fixed as intermediate level (i.e., the zero level of coded values), three-dimensional surface plots were made by using the surface characterizer of JMP10.0 software, and then orthographic projection was made (Figure 7).

The induction of protein gel formation by DPCD is the result of the combined action of the two effects. One is the thermal effect; under the induction of heat, myosin will gradually denaturate, stretch, and aggregate to form gel [23]. The other one is the molecular effect of CO_2_; at a high-pressure state, CO_2_ dissolves in water to form carbon acid, and carbon acid dissociates hydrogen ions, reducing the system’s pH. Hydrogen ions also make the myosin molecules positively charged. At the same time, CO_2_ is a non-polar molecule, which can interact with the hydrophobic group of proteins, and denaturate myosin through electrostatic interaction, hydrogen bond, hydrophobic interaction. Myosin molecules, in turn, cross-link each other through hydrophobic interactions, disulfide bonds, nondisulfide covalent bonds to form a three-dimensional meshwork gel [16,17].

As can be seen from Figure 7B, the gel strength of shrimp surimi increased rapidly under the condition of low temperature with the increase in pressure, while the gel strength of shrimp surimi slowly increased with increasing pressure at high temperature. This may be because, at low temperatures, CO_2_ molecules moved slowly, and increasing the pressure increased CO_2_ density, and the number of CO_2_ molecules that interact with the surimi increased. At the same time, the pressure increase also made CO_2_ more easily permeated and diffused into the surimi, and the effect of CO_2_ molecules was enhanced so that the gel strength increased rapidly [24].

It can be seen from Figure 7C that, under high pressure, the gel strength of shrimp surimi increased slowly with the increase in treatment temperature, while under low pressure, the gel strength of shrimp surimi increased rapidly with the increase in treatment temperature. The molecular effect of temperature on CO_2_ has two sides [16,17]. On the one hand, the increased temperature would aggravate the thermal movement of CO_2_ molecules (referred to as the thermal motion effect), which would benefit its penetration and diffusion into the surimi, and make sufficient interaction with myosin. On the other hand, the increasing temperature decreased CO_2_ density (referred to as the density drop effect), which was unfavorable for CO_2_ dissolution in surimi. Under the condition of high pressure, the density of CO_2_ was relatively high, and the molecular thermal motion effect caused by increasing temperature was slightly greater than the density drop effect, so the increase in temperature on gel strength was small. Under low-pressure conditions, the CO_2_ density was small, and the molecular thermal motion effect caused by increasing temperature was larger than the density drop effect, which would aggravate the molecular thermal motion of CO_2_ and favor its penetration and diffusion into surimi, thereby enhancing the gel strength.

It can be seen from Figure 7E,H that the gel strength of shrimp surimi increased gradually with increasing temperature and pressure regardless of long or short treatment time, which illustrated that both temperature and pressure were important factors affecting the gel strength of shrimp surimi. This may be because increasing the pressure increased the molecular effect of CO_2_, promoted myosin denaturation, and increased the strength of the shrimp surimi gel, regardless of the treatment time [13]. Regardless of treatment time, increasing temperature exacerbated CO_2_ molecular thermal movement, accelerated its penetration and diffusion into surimi, and thus enhanced the gel strength.

From Figure 7F,I, it can be seen that no matter high pressure or low pressure, whether high temperature and low temperature, surimi strength showed an obvious increasing trend with the treatment time, which was because the interaction between CO_2_ and myosin molecules was more sufficient with the extension of treatment time, thus enhancing the gel strength. In addition, it was also reported that DPCD treatment was more effective than heat treatment in inactivating cathepsin L activity that causes surimi gel deterioration. At the same time, heat treatment was more effective than DPCD treatment in inactivating TGase activity that enhances gel strength, which also promotes DPCD shrimp surimi gel strength to be stronger than heat-induced shrimp surimi [25].

### 3.4. Optimization and Verification of DPCD-Induced Gelation Process Parameters for Shrimp Surimi

The process parameters of DPCD-induced gel formation from shrimp surimi were optimized using the predicted descriptor function of JMP10.0 data processing software, and the results are presented in Figure 8. As can be seen from Figure 8, within the test range, when *x*_1_ = 1, *x*_2_ = 0, and *x*_3_ = 1, i.e., when the pressure was 30 MPa, the temperature was 55 °C and the time was 60 min, the DPCD-induced shrimp surimi reached a higher gel strength of 198.28 N mm.

In order to further verify the predictive reliability of regression model (6), three groups of tests were conducted, each of which was repeated three times. The test value and predicted value were tested by the average one-sample *t*-test (Table 8). The predicted value was the average value of the whole, and the test value was the sample. As illustrated in Table 8, there was no significant difference in gel strength between the experimental value and the predicted value of the model (*p* > 0.05). This further proves that regression model (6) can be used to predict the gel strength of shrimp surimi under DPCD induction at different pressure, temperature and time, optimize the process parameters, and analyze the influence of pressure, temperature and time on the gel strength of shrimp surimi. Since the variables in regression model (6) were coded values, the coded value formula in Table 1 was substituted into regression model (6), and the results were obtained as formula (7), which was the mathematical expression of the relationship between the strength (y) of the shrimp surimi gel and the actual variables pressure (*z*_1_), temperature (*z*_2_), and time (*z*_3_).
*y* = −1272.10 + 26.53*z*_1_ + 46.19*z*_2_ + 4.89*z*_3_ − 0.34*z*_1_*z*_2_ − 0.03*z*_1_*z*_3_ − 0.06*z*_2_*z*_3_ − 0.10*z*_1_^2^ − 0.30*z*_2_^2^(7)

### 3.5. Effect of DPCD on the Quality of Shrimp Surimi

It can be seen from Table 9 that compared with heat induction, DPCD reduced the loss of moisture, crude protein, and ash in the shrimp surimi gel. This may be because the preparation of shrimp surimi gel by heat induction was made by boiling water, during which part of the water-soluble proteins and minerals may dissolve in water, resulting in the loss of proteins and minerals in the gel. DPCD-induced shrimp surimi gels were carried out in an anhydrous CO_2_ environment. Although DPCD is somewhat extractable for moisture, it is not extractable for proteins and minerals, and the process is static, and only small amounts of water-soluble proteins and minerals may be carried out by moisture during unloading [26]. Whereas there was a small loss of protein from the gel and essentially no loss of minerals. Correspondingly, the weight loss of heat-induced shrimp surimi gel was 323.92% greater than that of DPCD.

Sensory evaluation showed that the sensory scores of DPCD-induced shrimp surimi gels were significantly higher than that of heat-induced (*p* < 0.05), which was mainly because heat induction was performed in a water bath, and more odor and taste substances were easily lost. In addition, the slow heat transfer rate and uneven mass transfer induced by heat also lead to the loose structure of the gel [27]. In contrast, DPCD induction was carried out in an anhydrous environment with less loss of taste substances. Although DPCD was extractable for volatile odorants, it was a static treatment and, therefore, only had a small loss during unloading pressure. The DPCD also denatured the protein sufficiently, which gave the shrimp surimi gelatin a bright pink color. In addition, the surface tension of CO_2_ in the supercritical state is zero, which made the dissolution and diffusion of CO_2_ in the shrimp surimi more uniform, thus forming a more uniform and dense gel morphology.

Gel strength and water-holding capacity of gels are important indicators for evaluating the quality of gels. As can be seen from Table 9, the gel strength and gel water-holding capacity of DPCD-induced shrimp surimi were significantly higher (*p* < 0.05) than those of heat-induced shrimp surimi gel. This is mainly because heat-resistant alkaline proteases (optimum 50–70 °C) present in shrimp surimi degraded myofibrillar proteins during heat-induced shrimp surimi gel, greatly reducing the elasticity and strength of the gel [3]. The temperature of DPCD to induce the formation of gels from surimi was below 55 °C; moreover, DPCD has a passivating effect on some high-temperature-resistant alkaline proteases [28]; Therefore, the gelation deterioration was inhibited. In addition, the gel strength is closely related to the microstructure of the gel. Generally speaking, the smaller the pore size and uniform distribution of the gel network, the bigger the gel strength and water-holding capacity of the gel will be. According to the sensory evaluation and microstructure observation of shrimp surimi gels, the gel network of heat-induced samples had large pore sizes and uneven distribution, while the gel network of DPCD-induced shrimp surimi gels had small pore sizes and uniform and compact distribution. Therefore, the gel strength and water-holding capacity of shrimp surimi induced by DPCD were relatively high.

The TPA properties of the shrimp surimi are shown in Figure 9B. The shrimp surimi gels were characterized by high chewiness, hardness, adhesiveness, springiness, and small cohesiveness. The chewiness, hardness, adhesiveness, and springiness of the DPCD-induced samples were significantly higher than that of the heat-induced ones, which corresponded to the gel strength results. This may be because DPCD promoted the formation of a fine and uniform gel network structure, which increased the water-holding capacity of the gel, so the texture properties of samples increased.

The low-field nuclear magnetic resonance (LF-NMR) curves (Figure 10) of shrimp surimi gels had four peaks whose relaxation times were denoted by *T*_2b1_, *T*_2b2_, *T*_21_, and *T*_22_, corresponding to different states of water in the gel, respectively. *T*_2b1_ and *T*_2b2_ (0–10 ms) represent bound water tightly bound to protein molecules. *T*_21_ (30–500 ms) represents immobilized water that entrapment by macromolecules inside the shrimp surimi gel. The *T*_22_ (500–2000 ms) represents free water in the shrimp surimi gel system [29]. The DPCD shrimp surimi gel lacked one *T*_22_ peak (i.e., free water) than the heat-induced gel, indicating that free water in DPCD was below the detectable limit. The relaxation time *T*_2_ is related to the binding force and degree of freedom of hydrogen protons in the sample. The larger the *T*_2_, the smaller the binding of water molecules and the greater the degrees of freedom [20].

As can be seen from Figure 9, *T*_2b1_, *T*_2b2,_ and *T*_21_ of the shrimp surimi gel induced by DPCD were significantly shorter than those of the heat-induced shrimp surimi gel (*p* < 0.05), indicating that DPCD increased the binding capacity of the shrimp surimi gel to bound water and immobilized water. *A*_2_ represents the relative content of water molecules in different states. It can be seen from Figure 9 that the immobilized water (*A*_21_) was the most important water in shrimp surimi, accounting for 89.93% and 93.55% of the heat-induced gel and DPCD gel, respectively. The relative contents of the bound water and immobilized water in the DPCD-induced gel were both higher than those in the heat-induced gel. It was further indicated that DPCD reduced the loss of bound water and immobilized water in shrimp surimi gel. This is closely related to the microstructure of shrimp surimi gel. It was known from sensory evaluation and microstructural analysis that heat-induced shrimp surimi gels had loose tissues with a large pore size of gel network, which easily made the transformation of immobilized water into free water for loss, while DPCD-induced shrimp surimi gels had dense tissues with uniform and small pore size of gel network, which locks immobilized water. This result corresponds to those of water-holding capacity, gel strength, and microstructure. It was also well illustrated that the shrimp surimi gel induced by DPCD was significantly superior to that induced by heat in terms of quality.

As shown in Figure 9A, the *L**, *a**, and *b** values of DPCD-induced shrimp surimi gel are significantly higher than those induced by heat (*p* < 0.05), which may be related to the microstructure of shrimp surimi gel formation. The shrimp surimi gel induced by DPCD is more uniform and dense than that induced by heat, which makes it more reflective to light, thus making its *L** value and *a** value correspondingly larger.

The appearance and SEM images of different shrimp surimi gels are shown in Figure 11. It can be seen from the appearance image that the gel structure of the heat-induced shrimp surimi was relatively loose (Figure 11A), and the surface of the product had the characteristics of large cracks and rough surfaces. The DPCD-induced shrimp surimi gel had a smooth surface and tight structure (Figure 11B). From the SEM images, it can be found that the holes in the network of heat-induced shrimp surimi gel were coarse, and the surface was bumpy. At the same time, DPCD gel had a fine network with fine and uniform pores. This further indicated that DPCD promoted the formation of a dense gel network structure and improved the gel quality of shrimp surimi. Rawdkuen, Sai-Ut, Khamsorn, Chaijan, and Benjakul [30] suggested that regular structure was intrinsic to stronger gel strength. Boiling heating in water usually caused the network pore size of surimi gels to be large and non-uniform due to the slow heating rate and non-uniform mass transfer [31]. During the treatment of DPCD, the surface tension of CO_2_ in the supercritical state was zero, which made its dissolution and diffusion in the sample relatively uniform, and it was beneficial to form a gel network structure with relatively small pore size and uniform and dense. Therefore, DPCD could induce shrimp surimi to form a gel with a better microstructure.

## 4. Conclusions

The mathematical model of gel strength and DPCD treatment temperature, pressure, and time of shrimp surimi was established by response surface methodology and regression analysis in the present study. DPCD treatment of pressure, temperature, and time and their interactions had significant effects on the gel strength of shrimp surimi, in which pressure had the greatest effect on the gel strength, followed by time and temperature. The DPCD-treated shrimp surimi reached a high gel strength (197.35 ± 2.02) N·mm when the DPCD treatment pressure was 30 MPa, the temperature was 55 °C, and the time was 60 min. Meanwhile, compared with the heat-induced shrimp surimi gel, DPCD reduced the nutrient loss and weight loss and increased the water-holding capacity and gel strength of the shrimp surimi gel. Therefore, the sensory score of DPCD-induced shrimp surimi gel was significantly higher than that of heat-induced shrimp surimi gel. DPCD increased the binding capacity of shrimp surimi to bound water and immobilized water and reduced their loss. The result of the microstructure showed that the pores of the gel network induced by DPCD were smaller and more uniform than those induced by heat. Overall, DPCD is a promising method that can be used to induce the formation of suitable gels from shrimp surimi. This study has established a prediction model between the temperature, pressure, and time of DPCD treatment and the gel strength of shrimp surimi. Producers can quickly determine the parameters of shrimp surimi gel according to the actual situation combined with this model.

## Figures and Tables

**Figure 1 foods-11-03807-f001:**
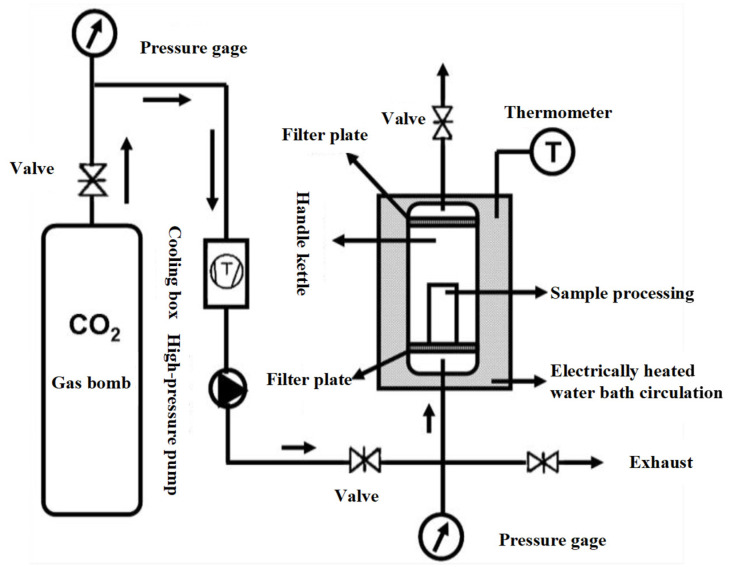
Process of dense phase carbon dioxide.

**Figure 2 foods-11-03807-f002:**
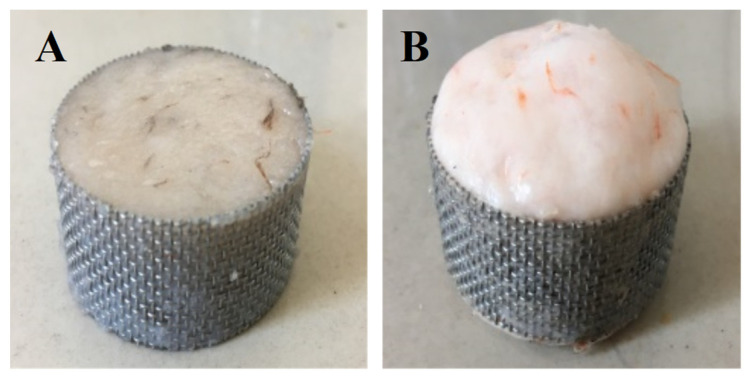
Shrimp surimi before DPCD treatment (**A**) and after DPCD treatment (**B**).

**Figure 3 foods-11-03807-f003:**
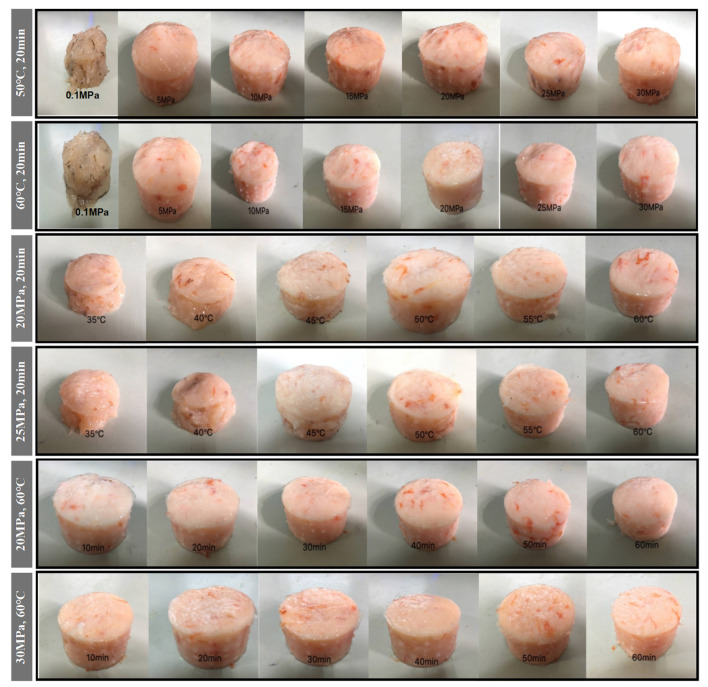
Visual appearance of shrimp surimi after different condition DPCD treatments.

**Figure 4 foods-11-03807-f004:**
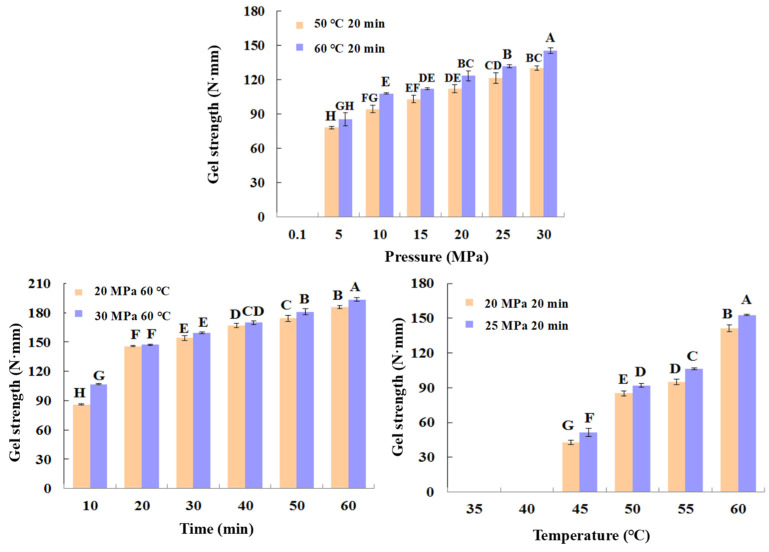
Gel strength (N·mm) of shrimp surimi after different condition DPCD treatments. Different capital letters indicate significant differences (*p* < 0.05).

**Figure 5 foods-11-03807-f005:**
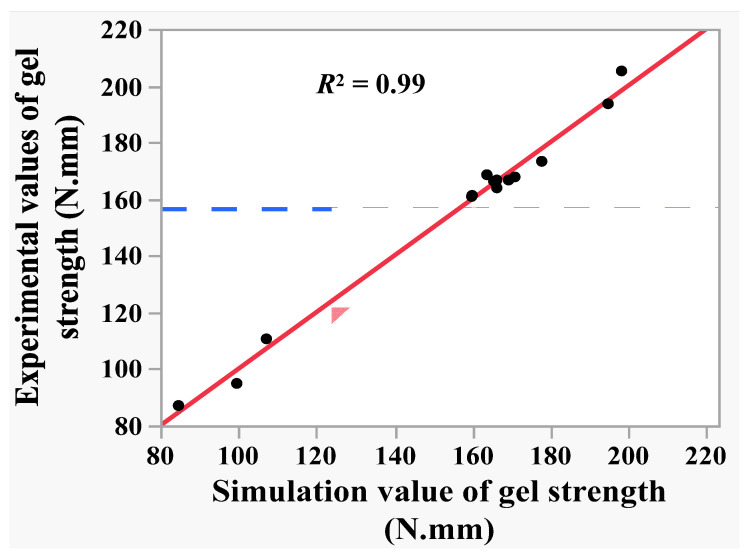
The correlation of experimental value and predicated value of gel strength.

**Figure 6 foods-11-03807-f006:**
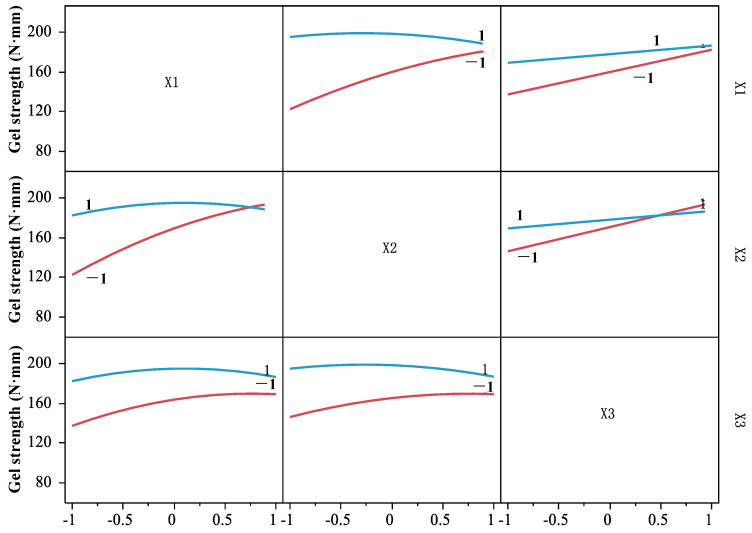
The interaction of pressure, temperature, and time.

**Figure 7 foods-11-03807-f007:**
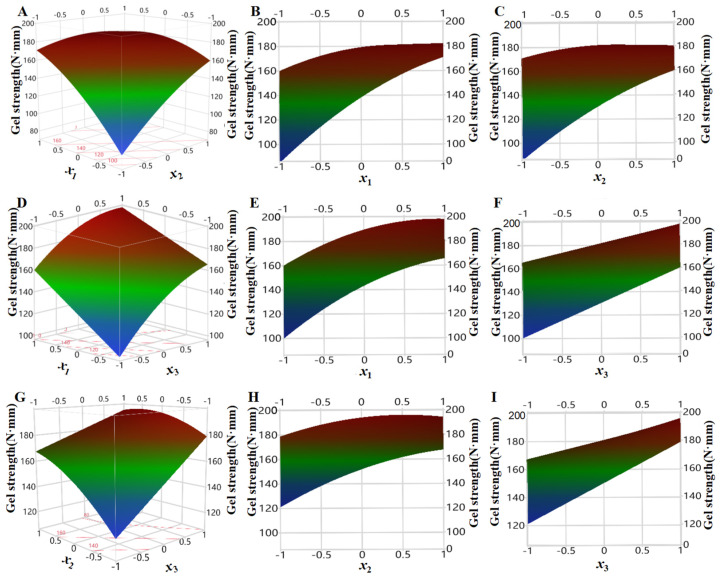
Effect of pressure, temperature, and time on the gel strength of shrimp surimi ((**A**): effect of pressure and temperature on gel strength; (**B**): effect of pressure on gel strength; (**C**): effect of temperature on gel strength; (**D**): effect of pressure and time on gel strength; (**E**): effect of pressure on gel strength; (**F**): effect of time on gel strength; (**G**): effect of temperature and time on gel strength; (**H**): effect of temperature on gel strength; (**I**): effect of time on gel strength).

**Figure 8 foods-11-03807-f008:**
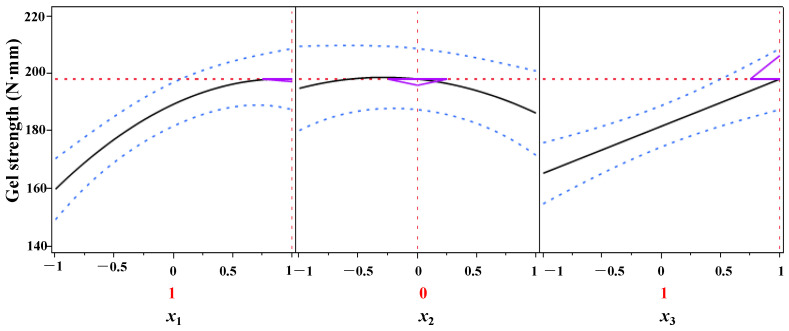
Optimization and prediction of the gel strength of shrimp surimi.

**Figure 9 foods-11-03807-f009:**
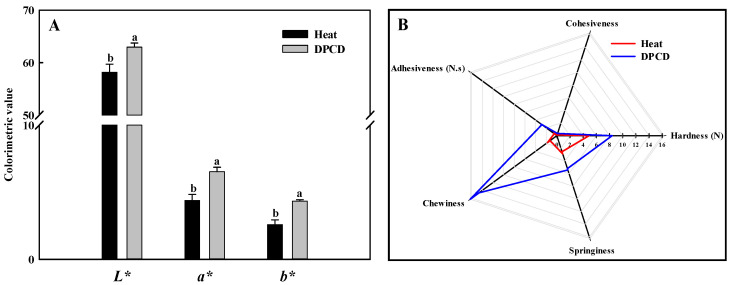
Color (**A**) and texture profile analysis (**B**) of shrimp surimi induced by heat and DPCD. Different lowercase letters indicate significant differences (*p* < 0.05).

**Figure 10 foods-11-03807-f010:**
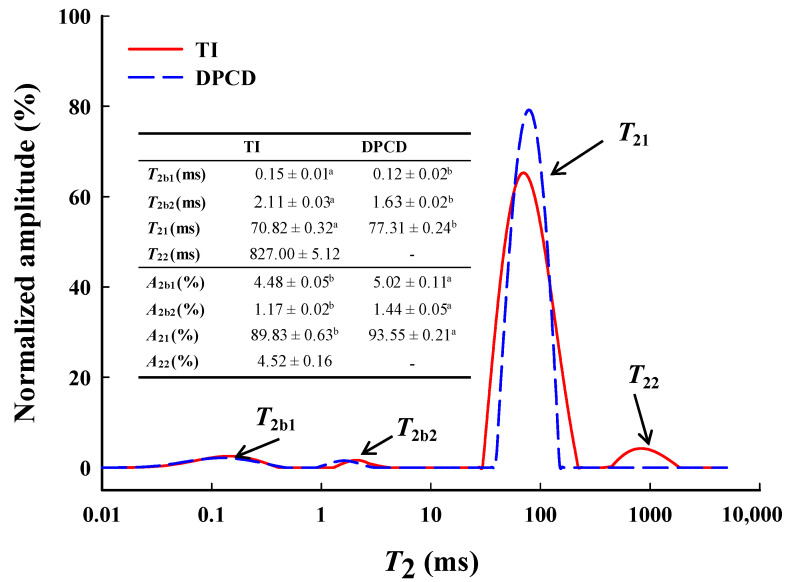
Low-field NMR curves, relaxation times *T*_2_ (ms), and amplitude area relative contents *A*_2_ (%) of shrimp surimi induced by heat and DPCD. Different lowercase letters indicate significance.

**Figure 11 foods-11-03807-f011:**
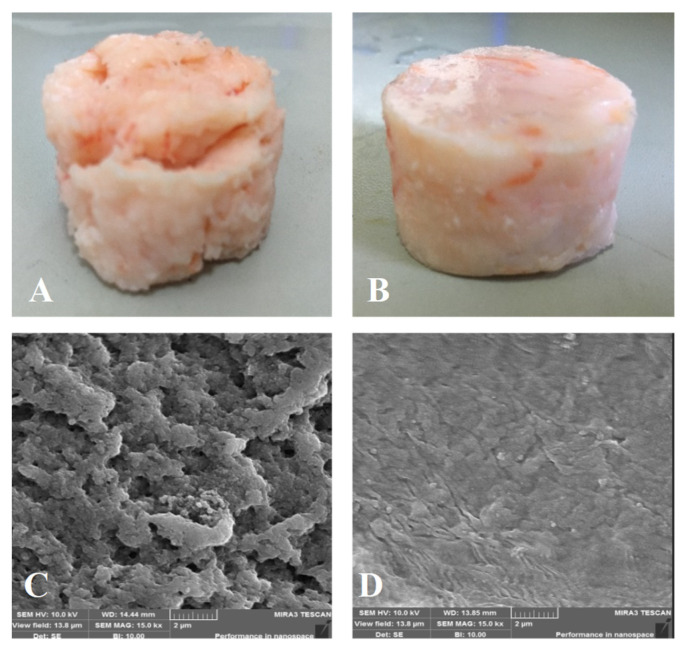
Gel appearance ((**A**): heat, (**B**): DPCD) and microstructure ((**C**): heat, (**D**): DPCD, ×1500) of shrimp surimi induced by heat and DPCD.

**Table 1 foods-11-03807-t001:** Coded values of factor and level.

Factors and Levels	Pressure (*z*_1_)/MPa	Temperature (*z*_2_)/°C	Time (*z*_3_)/min
1	30	60	60
0	20	55	35
−1	10	50	10
Δ*_j_*	10	5	25
Coded value formula	*x*_1_ = (*z*_1_ − 20)/10	*x*_2_ = (*z*_2_ − 55)/5	*x*_3_ = (*z*_3_ − 35)/25

**Table 2 foods-11-03807-t002:** Sensory evaluation standard of shrimp surimi gel.

Score	Odor	Color	Tissue Morphology	Taste
1	The fishy smell is relatively strong	Teal gray	The gel is soft and dense and cannot be cut	There is no umami flavor of shrimp, but there is a peculiar smell
2	Slightly fishy smell	Off white	The cut surface of the gel is looser, with a few uneven small pores; The gel breaks when pressed lightly with middle finger	Almost no shrimp meat umami
3	Bland shrimp aroma	Light pink	The cut surface of the gel is basically dense, with a few small air pores; The gel is concave without cracking when pressing the gel vigorously with middle finger, leaving the gel non-restoring	The umami taste is lighter, and the taste is normal
4	Shrimp aroma	Pink	The cut surface of the gel is dense, with a few small air pores; the gel is concave without cracking when pressed with the force of the middle finger, leaving the gel recovered	With the umami taste of shrimp, delicious and full of flavor
5	Tangy shrimp aroma	Bright pink	The cut surface of the gel is dense, with small and evenly distributed spiracles. The gel was markedly concave without cracking when slightly pressed against the middle finger, leaving the gel recovered	With the umami taste of shrimp, delicious and strong aftertaste

**Table 3 foods-11-03807-t003:** Scheme and result of Box-Behnken experimental design.

Number	Pressure (*x*_1_)	Temperature (*x*_2_)	Time (*x*_3_)	Gel Strength (*y*/N·mm)
1	10	50	35	87.07 ± 0.66
2	10	50	35	160.86 ± 1.02
3	30	60	35	167.74 ± 1.39
4	30	60	35	173.25 ± 1.92
5	20	50	10	110.60 ± 1.90
6	20	50	60	166.66 ± 0.44
7	20	60	10	168.52 ± 1.81
8	20	60	60	193.57 ± 1.60
9	10	55	10	94.87 ± 3.53
10	30	55	10	166.33 ± 1.17
11	10	55	60	161.26 ± 1.08
12	30	55	60	205.15 ± 0.21
13	20	55	35	163.97 ± 0.40
14	20	55	35	166.67 ± 1.41
15	20	55	35	163.90 ± 0.33

**Table 4 foods-11-03807-t004:** Model summary.

Std.De.	Mean	C.V.%	*R* ^2^	Adjusted *R*^2^	Predicted *R*^2^	Adeq Precision
5.36	156.69	3.42	0.99	0.97	0.85	26.46

**Table 5 foods-11-03807-t005:** Coefficients of regression model (5) and their significant test.

Item	Coefficient	Standard Error	*t* Value	*p*-Value
Intercept	164.85	3.09	53.29	<0.0001 *
*x* _1_	26.05	1.89	13.75	<0.0001 *
*x* _2_	20.52	1.89	10.83	<0.0001 *
*x* _3_	23.29	1.89	12.30	<0.0001 *
*x* _1_ *x* _2_	−17.07	2.68	−6.37	0.0014 *
*x* _1_ *x* _3_	−6.89	2.68	−2.57	0.0499 *
*x* _2_ *x* _3_	−7.75	2.68	−2.89	0.0340 *
*x* _1_ ^2^	−10.28	2.79	−3.69	0.0142 *
*x* _2_ ^2^	−7.34	2.79	−2.63	0.0464 *
*x* _3_ ^2^	2.33	2.79	0.84	0.4411

* means significant effect (*p* < 0.05).

**Table 6 foods-11-03807-t006:** Variance and lack of fit analysis of regression model.

Sources	D f	Sum of Squares	Mean Square	*F*	*p*
Model	9	15,309.33	1913.67	70.19	<0.0001 *
Residual	5	163.59	27.27		
Cor Total	14	15,472.92			
Lack of Fit	3	138.53	46.18	18.51	0.0517
Pure error	2	4.99	2.49		
Total error	6	163.59			
		*R*^2^ = 0.99	*R*_Adj_^2^ = 0.97		

* means significant difference (*p* < 0.05).

**Table 7 foods-11-03807-t007:** Coefficients of regression model (6) and their significant test.

Item	Coefficient	Standard Error	*t*	*p*
Intercept	166.28	2.51	66.29	<0.0001 *
*x* _1_	26.05	1.85	14.11	<0.0001 *
*x* _2_	20.52	1.85	11.11	<0.0001 *
*x* _3_	23.29	1.85	12.62	<0.0001 *
*x* _1_ *x* _2_	−17.07	2.61	−6.54	0.0006 *
*x* _1_ *x* _3_	−6.89	2.61	−2.64	0.0385 *
*x* _2_ *x* _3_	−7.75	2.61	−2.97	0.0250 *
*x* _1_ ^2^	−10.46	2.71	−3.86	0.0084 *
*x* _2_ ^2^	−7.52	2.71	−2.78	0.0322 *

Bulleted list * means significant effect (*p* < 0.05).

**Table 8 foods-11-03807-t008:** Verification test of regression model (6).

Pressure (MPa)	Temperature (°C)	Time (min)	Gel Strength Test Value (N·mm)	Gel Strength Prediction Value (N·mm)	Relative Error	*p*-Value of *t*-Test
30	55	60	197.35 ± 2.02	198.28	0.469%	0.51
25	50	20	139.46 ± 2.66	140.63	0.832%	0.52
20	55	40	169.45 ± 2.67	170.94	0.872%	0.44

**Table 9 foods-11-03807-t009:** Changes in gel quality of shrimp surimi induced by heat and DPCD.

		Heat	DPCD
Nutrient content	Water (%)	74.05 ± 1.23 ^b^	77.31 ± 0.93 ^a^
	Protein (%)	13.35 ± 0.98 ^b^	17.81 ± 0.58 ^a^
	Ash (%)	2.10 ± 0.06 ^b^	3.69 ± 0.19 ^a^
Weight loss (%)		18.78 ± 1.37 ^b^	4.43 ± 0.22 ^a^
Sensory score		4.63 + 1.21 ^b^	9.12 ± 1.13 ^a^
Water-holding capacity (%)		84.34 ± 2.82 ^b^	75.50 ± 1.37 ^a^
Gel strength (N·mm)		25.20 ± 3.15 ^b^	203.44 ± 3.69 ^a^

Different lowercase letters indicate significant differences (*p* < 0.05).

## Data Availability

Data is contained within the article.

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
