# Peer review of "Gelation Process Optimization of Shrimp Surimi Induced by Dense Phase Carbon Dioxide and Quality Evaluation of Gel"

_foods, 2022, doi:10.3390/foods11233807_

Round 1

Reviewer 1 Report

1.       Instrumental color like CIE L*, a*, b* is missing. Please provide these color parameters.

2.       Please explain the need to prepare "surimi" rather than minced shrimp meat.

3.       Line 88. Did you add salt when making the "sol"? Unless, why? Protein solubility is a prerequisite for gelation, as you are acquainted.

4.       Line 111-112. How long did it take the equilibrium to develop?

5.       Line 153. Check the heading.

6.       Why was gel strength presented instead of breaking force and deformation?

7.       How about the Texture Profile Analysis of the gel?

8.       Line 373-388. It is important to talk more about how DPCD affects the proteolysis of surimi as well as how it affects the remaining transglutaminase activity of surimi.

9.       In conclusion, if you have any recommendations for how this method might be used commercially.

10.   Fig. 8. Please provide the magnification of the image.

11.   Please expand on the impact of DPCD on lipid oxidation, astaxanthin pigment, and the microbiological quality of the resultant gels.

Reviewer 2 Report

Hi dear

This article "Gelation process optimization of shrimp surimi induced by dense phase carbon dioxide and quality evaluation of gel” was revised and has a novelty and I recommend it for major revision.

Title: It is perfect and complete.

Abstract:

·       Please include a background of the study. For example surimi and the aim of study.

·       The type and detail of statistical design used in this research should be mentioned.

Introduction:

·       Introduction is really perfect but it should be better if include the RSM characterization i.e., the repeat of factorial and axial and central points and α etc. in the end of introduction.

Materials:

·                 Please write materials as Company Name (City, Country), especially for chemical analysis assessment which used in the study.

Methodology:

·       Line 121: test target exchange to test response.

·       Table 1: What is the reason for choosing this domain of your invoices?

·       Line 132-138: The way of expressing the method of measuring macronutrients and other parameters has a scientific flaw. Please take help from the following article for the correct way of expressing it, so that the standard number of the working method should be clearly stated (https://doi.org/10.1590/fst.60820).

 Results:

·       Table 3: Please exchange code to real data for better understanding.

·       Line 199-200: In Formula 1: please pointe to R2, Adjuasted-R2, CV, and Adeq Precision.

·       Table 6: Please pay attention to more articles for a better understanding of my request and take ideas from them both in terms of the characteristics of the RSM plan and the types of tables and figures.

·       Fig 4: It should be design as 3D figure for better comprehension.

·       Table 7: Please have a statistical comparing between the test value and the prediction value.

·       Why was the basis for choosing the optimal sample based on gel strength and not other parameters such as color and sensory changes, weight loss, and water holding capacity?

Discussion:

Discussion text must grammar improve and in some cases it is very weak and maybe there is no discussion at all.

References: It is OK.

The article has many flaws in express and concept of English, it is suggested to be revised in a scientific and native way.

Round 2

Reviewer 1 Report

All points raised by reviewers were carefully addressed and answered point-by-point. So, it can be accepted.